# The older the injured, the worse the outcomes: A comparison of injury patterns and in-hospital outcomes between younger and older adult trauma patients at a tertiary hospital in Northern Tanzania

Edwin Joseph Shewiyo[1,2,3], Rosalia Njau [2,3], Natan Nascimento de Oliveira[3,4], Frijenia G. Sumbai[1,3], Paige O'Leary[3,4], Frida Shayo[5,1,2], João Vitor Perez Souza [3,4], João Ricardo Nickenig Vissoci[3,4], Blandina T. Mmbaga [5,1,2], Catherine A. Staton [3,4]*

1 School of Public Health, KCMC University, Moshi, Tanzania, 2 Kilimanjaro Clinical Research Institute, Moshi, Tanzania 3 Global Emergency Medicine Innovation and Implementation (GEMINI) Research Center, Duke Global Health Institute, Duke University, Durham, North Carolina, United States of America, 4 Duke Department of Emergency Medicine, Durham, North Carolina, United States of America. 5 Department of Emergency Medicine, Kilimanjaro Christian Medical Center (KCMC), Moshi, Tanzania

* catherine.lynch@duke.edu

## Abstract

The number of older adults (>60 years) in Sub-Saharan Africa (SSA) is expected to double by 2050. This demographic shift has led to a rise in traumatic injuries among this population, with one in ten trauma patients in Tanzania being an older adult. Yet, geriatric specialization remains largely absent in many low- and middle-income countries (LMICs), including Tanzania. To inform strategies for optimizing care in this vulnerable group, we conducted a cross-sectional secondary analysis using data from the adult trauma registry at Kilimanjaro Christian Medical Centre (KCMC), a tertiary hospital serving over 15 million people in northern Tanzania. The study included all injured adults (>18 years) from 2020 to 2024. We compared socio-demographic, clinical characteristics, injury patterns, and in-hospital outcomes between older (≥60 years) and younger adults. The main outcomes were length of hospital stay and in-hospital mortality. A total of 3,296 adult trauma patients were included, of whom 13.3% were older adults. Older adults took 4 hours longer to reach care (p < 0.001) and were more likely to be injured by falls (56% vs. 17%), while younger adults were more commonly involved in road traffic crashes (67% vs. 33%) (p < 0.001). Although most older adults sustained mild injuries (86%), they were more often hospitalized (91%) and required surgery (66%) compared to younger adults (85%, 58%) (p = 0.002). Comorbidities were more prevalent in older adults, notably diabetes (12%) and hypertension (26%) (p < 0.001). Older adults also experienced 4 days longer hospital stays and higher in-hospital mortality (9% vs. 4%) than younger adults (p < 0.001). Our study demonstrates significant differences between the injured

**Data availability statement:** Minimal data for this study are unable to be shared publicly due to a data sharing agreement with the Tanzanian government and regulatory ethical authorities of the country which only allows the authors to share data upon reasonable request, as patients did not consent to public data transfer. Requests require a written agreement approved by Kilimanjaro Christian Medical Centre (institutional) Ethics Committee and the National Institute for Medical Research (Tanzania) in protection of the patient's privacy. Data inquiries are readily approved and processed in a timely manner upon request and can be sent to Gwamaka W. Nelson via email (gwamakawilliam14@gmail.com) for researchers who meet the criteria for access to confidential data.

**Funding:** This study was financially supported by the National Institutes of Health (NIH) (https://www.nih.gov) and the Duke Department of Emergency Medicine (https://emergencymedicine.duke.edu), via the Trauma Research Capacity Building Program in KCMC D43 TRECK (https://sites.globalhealth.duke.edu/gemini/projects/treck-d43), in the form of a grant (D43-TW012205) received by BTM and CAS, and in the form of awards received by EJS, RN, and FGS. This study was also financially supported by the National Institutes of Health (NIH) in the form of a data trauma registry funded by the NIH PRACT via clinical trial at KCMC grant (R01-CAA027512). No additional external funding was received for this study. The funders had no role in study design, data collection and analysis, decision to publish, or preparation of the manuscript.

**Competing interests:** The authors have declared that no competing interests exist.

older adult patients and the younger adult patients, highlighting the differences in care required for older adult trauma patients and providing evidence to support the next steps of adoption and implementation of older adult-specific clinical practice guidelines to improve outcomes.

## Introduction

The global population of people aged 60 years and above is growing faster than all younger age groups. As of 2020, over 1 billion people were over 60 years of age and the population is projected to reach 2 billion by 2050 [1,2]. In the last 60 years, life expectancy has increased globally [3,4]. In Tanzania, life expectancy has increased from 43 to 66 years [5,6]. With this demographic shift comes an increased need to understand and address the unique healthcare needs of older adults, such as a shift from communicable diseases to noncommunicable diseases; from injuries caused by road traffic crashes to injuries increasingly caused by falls. Among the various challenges faced by this population, injuries pose a substantial threat to overall health and quality of life [3,4]. In Tanzania, approximately 11% of all injuries occur among older adults, who are more prone to post-injury disabilities [7–9]. As of late, people aged 60 and above make up ~6% of the population in Tanzania and ~10% in the Kilimanjaro region [10]. The impacts of this demographic shift need to be understood in order to update healthcare policies.

Currently, little is known about how older adult injury patients interact with healthcare systems in low-middle-income countries, such as Tanzania. Specific challenges faced by older adults along the trauma care continuum are unknown compared to the younger population in Tanzania. Understanding these challenges will allow for evidence-based practices to be put in place for this population. In high-income countries (HICs), clinical practice guidelines for older adult trauma patients have been implemented and reduced older adult injury patient mortality rates by up to 20% and shown 10% lower rates of complications following trauma [11–15]. Currently, there are no guidelines for older adult trauma patients in Tanzania. Specific care guidelines are effective because aging brings on unique healthcare needs such as physiological and anatomical changes, such as decreased bone density, muscle mass, and sensory perception, making older adults more vulnerable to injuries and poor outcomes compared to younger adults [16–18]. Older adults also have different lifestyles compared to younger adults exposing them to different kinds of illnesses and injuries. For example, injuries due to falls are more common and are associated with a higher likelihood of morbidity and mortality [19,20]. Currently, the healthcare characteristics of older adult trauma patients in Tanzania are unknown, which is crucial for developing effective interventions, such as guidelines, to optimize trauma care for the older adult population, which is rapidly growing in Tanzania.

This study aims to address the current gap in knowledge regarding the characteristics of older adult trauma patients in Tanzania. Specifically, we will assess these characteristics at Kilimanjaro Christian Medical Centre (KCMC), a zonal referral

hospital in Northern Tanzania that serves both as a clinical training site and research site for medical care for the region. Examining the difference between older adults and the younger population of injured patients at KCMC will determine the needs of older adults along the trauma care continuum. Common injuries among all adult patients at KCMC include brain trauma, spinal cord trauma, bone fractures, and burns [9,21], as well as road traffic crashes with reported high prevalence in the region [22]. As such, we aim to describe the characteristics, and in-hospital outcomes for older adult trauma patients compared to the younger population at KCMC. We hope this descriptive analysis sets the stage for a quality improvement process at KCMC to inform older adult trauma patient care strategies.

## Methods

### Study design and setting

This was a hospital-based cross-sectional secondary analysis of the ongoing KCMC hospital trauma registry. The KCMC hospital trauma registry is a prospective registry established in 2013 under an NIH grant for quality improvement purposes. It was initially established to evaluate quality of care for TBI (Traumatic Brain Injury) patients and to improve care at the Emergency Department. This was the first prospective TBI registry established in Sub Saharan Africa to evaluate quality of care at the Emergency Department [23]. The registry later matured in 2018 to include all trauma patients. KCMC is a tertiary, zonal referral hospital serving more than 15 million people in the Northern zone of Tanzania. The hospital has the capacity of 630 official beds, 90 canvases, and 40 baby incubators and attends more than 2000 trauma patients annually with about 10% of all trauma patients being older adult patients [9].

### Study population

This study used all adult trauma patients who presented for care at the KCMC Emergency Department from January 2020 to April 2024 who were enrolled in the trauma registry. For our analysis purposes, patients were divided into two groups: 1) 18–59 years and 2) 60 years and above.

### Variables

The primary outcomes measured in this study were length of hospital stay (LOHS) and in-hospital death. We defined LOHS as the number of hospital days from admission to hospital discharge or death.

The socio-demographic characteristics of the sample population included age (years), sex, education, marital status, and employment status. We modeled age as a numerical variable in years. Sex was categorized as male and female according to local culture. Education was categorized as having no formal education, completing primary education, or completing secondary/tertiary education. Marital status was categorized as with or without a partner. Employment status was categorized as unemployed/student, professional/skilled employment or unskilled employment.

Clinical characteristics included the patient's insurance status, comorbidities status (diabetes, hypertension, HIV), and alcohol use prior to injury. Insurance status was categorized as none/abscond for patients who absconded from hospital bills, insured for patients using national health insurance/other ("other" which includes all the patients who have any other insurance or who were exempted from paying any hospital bills through social welfare) and out-of-pocket payments. Diabetes, hypertension, and HIV status were self-reported during admission of the patient. We categorized diabetes, hypertension, and HIV comorbidities status if the patient was tested and negative (doesn't have the disease), tested and positive (has the disease), and never tested (or does not know). Alcohol use was positive if the patient self-reported alcohol use up to 6 hours prior to injury, if a clinician determined alcohol intoxication, or if the patient had a positive breathalyzer test on arrival to the hospital.

Injury characteristics included injury type, part of the body injured, mechanism of injury, triage nurse assessment of the patient, time to care, level of consciousness upon admission to the ED, and injury severity as measured by the Kampala

Trauma Score (KTS). Injury type was a variable indicating (yes/no) if a patient had a fracture, burn, laceration/abrasion, puncture/penetration, contusion/hematoma, and other (which includes all other unspecified injuries or non-physical injuries such as drowns). Part of the body injured was a variable indicating (yes/no) if a patient had sustained injury on the head, chest, abdomen, hip, long bones, short bones, or other parts of the body. Triage nurse assessment was a variable indicating (yes/no) if an injured patient needed hospitalization, needed surgery, or had life-threatening injuries. Mechanism of injury was categorized as road traffic injury (RTI), fall, and other (burn, blunt, penetrating injuries, and drowning). Time to care was a continuous variable measured as the number of hours spent from injury to reaching care at the hospital. Injury severity was categorized as mild and moderate/severe based on the Kampala Trauma Score (KTS). A score of 9–10 was considered mild injury and a score of 0–8 was considered to be moderate to severe injury. The Kampala Trauma Score (KTS) is a validated tool that requires minimal data and has been widely used to determine injury severity in low-resource settings. It uses information on age, systolic blood pressure, respiratory rate, neurological status, and number of injuries [24,25]. The level of consciousness was measured from the Glasgow Coma Scale (GCS). The scale is widely used to objectively describe the extent of impaired consciousness in all types of acute medical and trauma patients. The scale assesses patients according to three aspects of responsiveness: eye-opening with a score of 4, motor with a score of 6, and verbal responses with a score of 5 with a total score of 15 [26]. The GCS was categorized as mild (for scores of 13–15), moderate (for scores of 9–12), and severely impaired level of consciousness (for scores of 3–8).

### Data analysis

Data analysis was done using R Software for Statistical Computing version 4.3 [27]. We computed descriptive statistics for all variables stratified by age, comparing characteristics between older and younger adult injury patients. We used a complete case analysis approach in our descriptive analysis after establishing that missingness was at random by visualization and proportions of missingness of the variables. The number of missing observations was reported for each variable on the tables. Continuous variables were presented as median and interquartile ranges. Categorical variables were presented as frequencies and percentages. For continuous variables, the Wilcoxon rank sum test was used to identify significant differences in medians between the older and younger adult patient groups. The test is suitable for comparison of a continuous variable between independent groups when the assumption of normality has not been met. For categorical variables, chi-square or Fisher's exact tests were used to identify significant differences in the distribution of observations between the younger adult and older adult patient groups. Based on the data we considered a significance level at 5% ($P < 0.05$).

### Ethical considerations

This study was reviewed and approved by the KCMC's ethics committee. Participant consent was waived as this study is a secondary analysis of a de-identified KCMC trauma registry data which is established for quality improvement.

## Results

### Enrollment and demographic characteristics of patients

We enrolled 3,296 eligible patients from the KCMC trauma registry, between January 2020 and April 2024. The median age of the sample was 34 years old (IQR 26, 48); younger adults represented 86.7% of the patients (median age 31; IQR 25, 41), while older adults were 13.3% of the enrolled subjects (median age 70; IQR 64, 79).

Table 1 shows the overall demographic sample and the distribution compared by age groups. Older adult patients presented significantly different demographic characteristics than younger adults (p < 0.001). While males were the majority of patients in the overall sample and in both age groups, older adults had a larger percentage of females than younger adults (43% and 16%, respectively). The Chagga tribe was more represented in older adults (63%), along with partnered

**Table 1. Demographic characteristics of enrolled patients stratified by age group (N = 3,296).**

| Variables | | Overall Sample | Age Group | | P-value[1] |
|---|---|---|---|---|---|
| | | | Younger Adult, N = 2,853 | Older Adult, N = 443 | |
| **Sex*** | Female | 654 (20%) | 464 (16%) | 190 (43%) | **<0.001** |
| | Male | 2,640 (80%) | 2,387 (84%) | 253 (57%) | |
| | Missing | 2 | 2 | 0 | |
| **Tribe*** | Chagga | 1,541 (47%) | 1,263 (44%) | 278 (63%) | **<0.001** |
| | Pare | 472 (14%) | 102 (3.6%) | 11 (2.5%) | |
| | Maasai | 166 (5.0%) | 155 (5.4%) | 11 (2.5%) | |
| | Other tribes | 1,115 (34%) | 1,030 (36%) | 85 (19%) | |
| | Missing | 2 | 2 | 0 | |
| **Marital Status*** | Without partner | 1,561 (47%) | 1,386 (49%) | 175 (40%) | **<0.001** |
| | With partner | 1,730 (53%) | 1,463 (51%) | 267 (60%) | |
| | Missing | 5 | 4 | 1 | |
| **Education*** | Primary education | 1,814 (56%) | 1,490 (53%) | 324 (75%) | **<0.001** |
| | Secondary or higher | 1,434 (44%) | 1,326 (47%) | 108 (25%) | |
| | Missing | 48 | 37 | 11 | |
| **Employment status*** | Student | 142 (4.3%) | 142 (5.0%) | 0 (0%) | **<0.001** |
| | Unemployed | 92 (2.8%) | 66 (2.3%) | 26 (5.9%) | |
| | Professional/Skilled | 1,311 (40%) | 1,248 (44%) | 63 (14%) | |
| | Unskilled | 1,672 (51%) | 1,333 (47%) | 339 (77%) | |
| | Other | 70 (2.1%) | 56 (2.0%) | 14 (3.2%) | |
| | Missing | 9 | 8 | 1 | |

*Variable with missing data

[1]Pearson's chi-square test or Fisher's exact test

individuals (60%) and those with lower formal education levels (primary education only, 75%) than younger adults. Considering the employment status, most older adults were unskilled employees (77%), and presented a higher rate of unemployment (5.9%) than younger adults (2.3%).

## Clinical characteristics of patients

Table 2 presents the clinical characteristics of the overall sample and the clinical characteristics compared by age group. Overall, the majority of injury patients pay out of pocket for hospital bills (80%), with older adults being more covered by health insurance than younger adults (39% vs 17%) (p < 0.001). The diagnosis of NCDs, such as hypertension and diabetes, was higher in older adults (26 and 12%, respectively) than younger adults (2% and 1% respectively) (p < 0.001); however, while the HIV prevalence was the same in both age groups (2.9%), there were higher proportions of older adults who never screened for HIV than younger adults (38% vs 27%) (p < 0.001). The use of alcohol prior to the injury was higher among younger adults (22%) than older adults (15%) (p = 0.001).

## Injury characteristics of patients

The most common type of injury was fracture (60%) and the most affected body part was the long bones (47%). However, when stratified by age, the fractures represented (75%) of the injuries presented by older adults (vs 57% of younger adults) (p < 0.001), and the long bones were affected in (67%) of the older patients (vs 43% of younger adults) (p < 0.001). The main mechanism of injury for younger adults was road traffic accidents (67%), while for older adults, falls accounted

**Table 2. Clinical and health-related characteristics of patients stratified by age group (N = 3,296).**

| Variables | | Overall Sample | Age Group | | P-value[1] |
|---|---|---|---|---|---|
| | | | Younger Adult, N = 2,853 | Older Adult, N = 443 | |
| **Insurance*** | None/Abscond | 7 (0.2%) | 7 (0.2%) | 0 (0%) | **<0.001** |
| | Out of pocket | 2,640 (80%) | 2,371 (83%) | 269 (61%) | |
| | Insured | 642 (20%) | 470 (17%) | 172 (39%) | |
| | Missing | 7 | 5 | 2 | |
| **Diabetes*** | No | 2,878 (88%) | 2,504 (88%) | 374 (84%) | **<0.001** |
| | Yes | 94 (2.9%) | 40 (1.4%) | 54 (12%) | |
| | Don't know | 316 (9.6%) | 301 (11%) | 15 (3.4%) | |
| | Missing | 8 | 8 | | |
| **Hypertension*** | No | 2,808 (85%) | 2,500 (88%) | 308 (70%) | **<0.001** |
| | Yes | 180 (5.5%) | 65 (2.3%) | 115 (26%) | |
| | Don't know | 299 (9.1%) | 280 (9.8%) | 19 (4.3%) | |
| | Missing | 9 | 8 | 1 | |
| **HIV Infection*** | Negative | 2,118 (69%) | 1,876 (70%) | 242 (59%) | **<0.001** |
| | Positive | 88 (2.9%) | 76 (2.9%) | 12 (2.9%) | |
| | Never tested | 871 (28%) | 713 (27%) | 158 (38%) | |
| | Missing | 219 | 188 | 31 | |
| **Alcohol use*** | No | 2,494 (79%) | 2,130 (78%) | 364 (85%) | **0.001** |
| | Yes | 671 (21%) | 605 (22%) | 66 (15%) | |
| | Missing | 131 | 118 | 13 | |

*Variable with missing data

[1]Pearson's chi-square test or Fisher's exact test

for the majority of the injuries (56%) (p < 0.001). The time interval between the injury and the arrival at the ED was about 4 hours longer in older adults than younger adults (Median 10.7 hours vs 6.3 hours) (p < 0.001). Both the Level of consciousness on admission (GCS) and the Injury Severity (KTS) were less severe in older adults than in younger adults. During triaging, There were higher proportions of older who needed hospitalization (91%), required surgeries (66%), or had life-threatening conditions (29%) than younger adults (85%, 58%, 21% respectively) (see Table 3).

### In-hospital outcomes of patients

The outcomes of older patients were poorer than younger adults (p < 0.001), as shown in Table 4: 9% of older adults died in the hospital (vs 4% of younger adults). Older adults stayed 4 days longer in the hospital until discharge than younger adults (median 9 days vs 5 days).

### Discussion

To the best of our knowledge, this is the first analysis comparing older and younger adult injury patient characteristics in Tanzania, to determine the characteristics of the older adult trauma population. Our findings revealed that there are a significant portion of injured patients are older adults, at the Kilimanjaro Christian Medical Center. These older adults had significantly different injury characteristics compared to younger adults, such as clinical characteristics, in-hospital outcomes, and the primary mechanism of injury. Importantly, in-hospital outcomes for older trauma patients were worse while predictive scores (KTS and GCS) were less severe. Lastly, the older adult trauma patients had a significant delay in reaching care, more likely to need surgery and more comorbidities, likely contributing to longer hospital stays, and higher

**Table 3. Injury characteristics of patients stratified by age group (N = 3,296).**

| Variables | Overall Sample | Age Group | | P-value[1] |
|---|---|---|---|---|
| | | Younger Adult, N = 2,853 | Older Adult, N = 443 | |
| **Injury type (Yes/No)** | | | | |
| Fracture | 1,968 (60%) | 1,634 (57%) | 334 (75%) | **<0.001** |
| Burn | 45 (1.4%) | 37 (1.3%) | 8 (1.8%) | 0.390 |
| Laceration/Abrasion | 368 (11%) | 342 (12%) | 26 (5.9%) | **<0.001** |
| Puncture/Penetration | 182 (5.5%) | 164 (5.7%) | 18 (4.1%) | 0.149 |
| Contusion/Hematoma | 444 (13%) | 416 (15%) | 28 (6.3%) | **<0.001** |
| Other type of injury | 912 (28%) | 833 (29%) | 79 (18%) | **<0.001** |
| **Part of the body injured (Yes/No)** | | | | |
| Head | 1,336 (41%) | 1,248 (44%) | 88 (20%) | **<0.001** |
| Chest | 254 (7.7%) | 221 (7.7%) | 33 (7.4%) | 0.827 |
| Abdomen | 124 (3.8%) | 113 (4.0%) | 11 (2.5%) | 0.128 |
| Hip | 97 (2.9%) | 80 (2.8%) | 17 (3.8%) | 0.231 |
| Long Bones | 1,533 (47%) | 1,234 (43%) | 299 (67%) | **<0.001** |
| Short Bones | 445 (14%) | 408 (14%) | 37 (8.4%) | **<0.001** |
| Other parts of the body | 646 (20%) | 580 (20%) | 66 (15%) | **0.007** |
| **Triage nurse assessment (Yes/No)** | | | | |
| Need of hospitalization | 2,756 (86%) | 2,366 (85%) | 390 (91%) | **0.002** |
| Need Surgery | 1,886 (59%) | 1,603 (58%) | 283 (66%) | **0.002** |
| Life-threatening condition | 717 (22%) | 592 (21%) | 125 (29%) | **<0.001** |
| **Mechanism of Injury** | | | | |
| Road Traffic Injury | 2,070 (63%) | 1,925 (67%) | 145 (33%) | **<0.001** |
| Fall | 654 (20%) | 405 (14%) | 249 (56%) | |
| Blunt Force/Stuck/Hit | 338 (10%) | 311 (11%) | 27 (6%) | |
| Other | 234 (7.1%) | 212 (7%) | 22 (5%) | |
| **Time from injury to admission (hours)*** | | | | |
| Median (IQR) | 6.8 (2.9,20.5) | 6.3 (2.7,19.5) | 10.7 (4.5, 25.8) | **<0.001** |
| Missing | 39 | 34 | 5 | |
| **Level of Consciousness (GCS)*** | | | | |
| Mild (13–15) | 3,122 (95%) | 2,690 (95%) | 432 (98%) | **0.014** |
| Moderate (9–12) | 88 (2.7%) | 84 (3.0%) | 4 (0.9%) | |
| Severe (3–8) | 68 (2.1%) | 63 (2.2%) | 5 (1.1%) | |
| Missing | 18 | 16 | 2 | |
| **Injury severity (KTS)*** | | | | |
| Mild | 967 (35%) | 646 (27%) | 321 (86%) | <0.001 |
| Moderate/Severe (0–8) | 1,776 (65%) | 1,713 (73%) | 53 (14%) | |
| Missing | 563 | 494 | 69 | |

*Variable with missing data

[1]Pearson's chi-square test or Fisher's exact test; Wilcoxon rank sum test

mortality compared to younger adults. Our findings suggest that the older adult injury population needs a specialized focus and clinical support to improve outcomes.

The proportion of older adult injury patients in our study was higher compared to previous studies in Tanzania. Our study found that 13.3% of the injury population at KCMC was older than 60 years of age, compared to previous studies

**Table 4. In-hospital outcomes of patients stratified by age group (N = 3,296).**

| Variables | | Overall Sample | Age Group | | P-value |
|---|---|---|---|---|---|
| | | | Adult, N = 2,853 | Elderly, N = 443 | |
| Outcomes | | | | | |
| Outcome* | Discharged | 3,067 (95%) | 2,675 (96%) | 392 (91%) | **<0.001** |
| | Death | 145 (4.5%) | 107 (3.8%) | 38 (9%) | |
| | Missing | 84 | 71 | 13 | |
| Length of Hospital stay (days)* | Median (IQR) | 5.0 (1.0,14.0) | 5.0 (1.0,13.0) | 9.0 (3.0,20.0) | **<0.001** |
| | Missing | 90 | 74 | 16 | |

*Variable with missing data

1 Pearson's chi-square test or Fisher's exact test; Wilcoxon rank sum test

which had around 9%, from different regions in Tanzania [7–9]. In studies conducted in other areas of Africa, the older adult injury population made up less than 5% of their injury population [28–30]. KCMC's larger older adult injury population may be due to the demographic shift occurring globally. Currently, in high-income countries, such as Asia, Europe, and the Americas, the proportion of older adult injury patients is 20% or greater [18,31–34]. Globally, the older adult population is projected to double by 2050 [1,35] and our study findings of an increased older adult injury population, reflect this demographic shift. In response to the increase in older adults seeking care, HICs have made efforts to standardize care for older adult populations such as establishing protocols and specialized clinical practice guidelines for geriatric trauma care. These specialized guidelines have improved patient outcomes for this population [36–39]. The increase in older adults seeking care at KCMC poses an opportunity in a low-resource setting to adapt similar strategies that have been successful in HICs to optimize care among older adults at KCMC.

In addition, to the high proportion of older adults among our trauma patients, these older adults also had significantly different injury characteristics compared to younger adults in our study such as mechanism of injury, hospitalization, and surgery. Falls were the predominant mechanism of injury among older adults, accounting for over half of the injuries (56%) compared to only 14% in younger patients. This is consistent with studies from other settings, which reflected a high incidence of falls among older adults leading to care-seeking behavior [19,20,40]. The reason for the higher occurrence of falls among older adults is well understood, as people age their percentage of muscle decreases which negatively impacts their muscle strength, balance, and vision; which all increase the likelihood of falls [20,41,42]. Falls as the primary injury mechanism among our older adult population, may explain the higher proportion of females within our population. As women age they undergo menopause and have a higher rate of osteoporosis, which increases risk of injury during fall [31]. Targeted fall prevention strategies, including home safety assessments, exercise programs, and medication reviews, are important for reducing fall-related injuries among older adults [41–44]. Among the older adult injury population 88% were hospitalized, and 64% required surgery. This finding shows the increased demand for the healthcare system required to care for older adults, which is consistent with other studies that demonstrated high costs and resources required in older adult trauma care [45–47]. Additionally, our study showed that the majority (60%) of older adults were paying out of pocket for their hospital bills and more than 80% had limited income due to their age and working abilities. Addressing the burden of older adult injury care on the hospital and at the individual and family level is critical, and will require specific solutions for this population.

Lastly, the older adult trauma patients had a significant delay in reaching care, likely contributing to longer hospital stays, and higher mortality compared to younger adults. Our study found that it took older adults twice as long to reach care following injury compared to younger adults. Older adults may have experienced this delay because of the referral system and lack of a proper support system to assist them to care following injury [48–50]. This delay in accessing care likely contributes to the higher morbidity and mortality observed in the older population evident in our study and others [51–53]. In our study,

we also found that older adults were staying at the hospital for 4 days longer compared to younger adults. Additionally, 1 out of 10 older adults died following injuries which was twice as high compared to younger adults which is consistent with previous studies [7,8,20,40,54]. The longer hospital stays and higher mortality in older adults may be due to the complexities of older adult injury impacted by comorbidities. Within our population, older adults had a significantly higher rate of diabetes and hypertension compared to younger adults, which impacted care provisions [17,31]. Frailty due to aging may have also impacted recovery and increased the risk of complications in our population, as frailty has been shown to negatively impact health outcomes by 2–5 times [15,36,37,39]. Comprehensive geriatric assessment and tailored treatment approaches are essential for optimizing outcomes in older adult trauma patients given they already have delays in reaching care, stay longer in hospitals following injuries, and are more likely to die compared to younger adults [54–56].

Our study has both strengths and limitations. Our study's demographic relevance is a significant strength, addressing the growing global and local population of older adults. The focus on older adults is crucial given the demographic transition and the associated healthcare challenges. Conducting the research at Kilimanjaro Christian Medical Centre (KCMC), which is a key referral hospital in Northern Tanzania, provides valuable insights specific to the region, enabling local healthcare policies and interventions. Additionally, the comparative analysis between older and younger adults highlights the unique needs of the older population. The findings of our study allow for a clear understanding of the differences in injury patterns and outcomes within this population that can directly influence hospital practices and protocols. Our study has several limitations, including the single-site nature of the study limits the generalizability of our findings. However, KCMC is the referral hospital for all of northern Tanzania, so this site is ideal to begin to understand the trauma occurring among older adults in this region. Additionally, long-term outcomes of these older adult trauma patients were not assessed, which would enhance the understanding of patient care. Lastly, differences in the severity of injuries between older and younger adults might also confound the results, this variability could impact the comparison of outcomes.

## Conclusion

Older adult trauma patients at KCMC have significantly different clinical characteristics such as in-hospital outcomes and primary mechanism of injury. They also have higher comorbid conditions, a significant delay in reaching care, longer hospital stays, higher surgical requirements, and higher mortality compared to younger adults. These findings highlight the patient, family, and health system burden that older adult trauma causes. To address this, The next steps are studies to adoption and implementation of a context-relevant evidence-based clinical practice guideline for older adults is needed to meet the unique clinical needs and to improve health outcomes in our low-resource setting for this population.

## Author contributions

**Conceptualization:** Edwin Joseph Shewiyo, Rosalia Njau, Natan Nascimento de Oliveira, Frijenia G. Sumbai, Paige O'Leary, Frida Shayo, João Vitor Perez Souza, João Ricardo Nickenig Vissoci, Catherine A. Staton.

**Formal analysis:** Edwin Joseph Shewiyo, Natan Nascimento de Oliveira, João Vitor Perez Souza, João Ricardo Nickenig Vissoci.

**Methodology:** Edwin Joseph Shewiyo, Rosalia Njau, Natan Nascimento de Oliveira, Frijenia G. Sumbai, João Vitor Perez Souza, João Ricardo Nickenig Vissoci.

**Project administration:** Blandina T. Mmbaga, Catherine A. Staton.

**Supervision:** Paige O'Leary, João Vitor Perez Souza, João Ricardo Nickenig Vissoci, Blandina T. Mmbaga, Catherine A. Staton.

**Writing – original draft:** Edwin Joseph Shewiyo, Rosalia Njau, Natan Nascimento de Oliveira, Frijenia G. Sumbai.

**Writing – review & editing:** Paige O'Leary, Frida Shayo, João Vitor Perez Souza, João Ricardo Nickenig Vissoci, Blandina T. Mmbaga, Catherine A. Staton.

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
