## [Decision Letter · Decision Letter 0]

10 Jan 2025

PGPH-D-24-02377

The older the injured, the worse the outcomes: A comparison of injury patterns and in-hospital outcomes between younger and older adult trauma patients at a tertiary hospital in Northern Tanzania.

Dear Dr. Staton,

Thank you for submitting your manuscript to PLOS Global Public Health. After careful consideration, we feel that it has merit but does not fully meet PLOS Global Public Health’s publication criteria as it currently stands. Therefore, we invite you to submit a revised version of the manuscript that addresses the points raised during the review process.

We look forward to receiving your revised manuscript.

Kind regards,

Hani Mowafi, M.D., M.P.H.

Academic Editor

Journal Requirements:

Additional Editor Comments (if provided):

Reviewers' comments:

Reviewer's Responses to Questions

**Comments to the Author**

1. Does this manuscript meet PLOS Global Public Health’s publication criteria?

Reviewer #1: Yes

Reviewer #2: Yes

2. Has the statistical analysis been performed appropriately and rigorously?

Reviewer #1: Yes

Reviewer #2: Yes

3. Have the authors made all data underlying the findings in their manuscript fully available (please refer to the Data Availability Statement at the start of the manuscript PDF file)?

Reviewer #1: No

Reviewer #2: Yes

4. Is the manuscript presented in an intelligible fashion and written in standard English?

Reviewer #1: Yes

Reviewer #2: No

Reviewer #1: 1. In the research objectives, your study addresses an important issue of trauma care in Tanzania’s aging population. However, it could benefit from explicitly linking the findings to actionable healthcare policies or interventions, which is currently missing...

2. In the methodology, using secondary data from the trauma registry is a good approach, but it lacks details about how missing or incomplete data were handled. Adding this would clarify potential biases and enhance transparency.

3. The statistical analysis is thorough and appropriate, especially with multivariable models for mortality. However, the manuscript doesn’t explain why certain statistical tests were chosen or how missing data were addressed, including these would help strengthen the analysis.

4. When comparing your findings to existing literature, you reference high-income countries well, while there are some references from sub-Saharan Africa but the manuscript lacks sufficient studies from sub-Saharan Africa or other LMICs. Including these would make your findings more regionally relevant.

5. The policy implications are strong, particularly the recommendation for geriatric-specific guidelines. To make it actionable, it would help to include examples of how guidelines from high-income countries could be adapted to Tanzania, which is currently missing.

6. The abstract is concise and informative, but it would be more impactful if it included a key policy takeaway or actionable recommendation, which is absent right now.

7. The introduction sets the stage well, but it doesn’t sharply highlight the specific gaps in Tanzanian healthcare policies for older adults. A clearer framing of these gaps could strengthen the narrative.

8. The results section is comprehensive, but the inclusion of numerous detailed tables can overwhelm readers. Moving less critical tables to supplementary materials could help streamline the presentation.

9. The tables and figures are clear, but the text often repeats data already presented visually. Summarizing key takeaways instead would make the narrative more concise and engaging.

10. The conclusion emphasizes the need for tailored guidelines, which is excellent. However, it would benefit from a stronger call to action, such as training programs or resource allocation, which is currently missing.

11. The references are solid, but there’s a lack of studies from sub-Saharan Africa or LMICs. Adding these would enhance the contextual relevance of the manuscript.

12. The authors should consider making the de-identified dataset used in the study available on a public platform, such as GitHub, Dryad, or Figshare. This would enhance the transparency, reproducibility, and impact of the research, allowing other researchers to build upon these findings. Highly recommend doing so... This would also comply with the PLOS Data Policy which requires “PLOS journals require authors to make all data necessary to replicate their study’s findings publicly available without restriction at the time of publication.”

Reviewer #2: In this research article, the authors compared demographic and clinical features of two groups of adults traumatized patients (18-60 years and >60 years of age) treated between 2020-2024 at a trauma hospital in Tanzania using prospectively collected registry data. Primary outcomes included length of hospital stay and in-hospital mortality. They found significant differences between the two groups, which may be useful to optimize care pathways of the older population.

In my view, the article can be of interest for the readership of Plos Global Health but requires some adjustments:

Abstract:

An analysis of in-hospital complications is anticipated in the 1st paragraph, but this is not described in the results section.

What is NCD?

Introduction:

The main research question is well summarized and overall well written

There are some un-necessary repetitions: “As of late, people aged 60 and above make up ~6% of the population in Tanzania and ~10% in the Kilimanjaro region” AND “In the Kilimanjaro region people aged 60 and above make up (10.4%) of the population, which…”

Methods

What type of registry was used? Some details about it?

No need to repeat two times the time interval

This study used ALL trauma patients who presented, should be corrected to ALL ADULTS…

"We defined in-hospital death as the death of a patient at any time after

admission to the hospital and as a reason for discharge”: what does this mean?

It should be mentioned that also the site of injury was analyzed and not only the type of injury.

Results, discussion, conclusions:

The conclusions are supported by the results and adequate. Limitations are clearly stated.

General comments:

A revision of the English language is necessary, including less verbosity in sentences.

The explanation of abbreviations don’t need to be repeated along all the article: once described at the beginning, just use the abbreviation in all the article.

**Do you want your identity to be public for this peer review?** For information about this choice, including consent withdrawal, please see our Privacy Policy

Reviewer #1: No

Reviewer #2: No

---

## [Decision Letter · Decision Letter 1]

7 Apr 2025

The older the injured, the worse the outcomes: A comparison of injury patterns and in-hospital outcomes between younger and older adult trauma patients at a tertiary hospital in Northern Tanzania.

PGPH-D-24-02377R1

Dear Dr. Staton,

We are pleased to inform you that your manuscript 'The older the injured, the worse the outcomes: A comparison of injury patterns and in-hospital outcomes between younger and older adult trauma patients at a tertiary hospital in Northern Tanzania.' has been provisionally accepted for publication in PLOS Global Public Health.

Best regards,

Hani Mowafi, M.D., M.P.H.

Academic Editor

Reviewer Comments (if any, and for reference):

Reviewer's Responses to Questions

**Comments to the Author**

Reviewer #1: All comments have been addressed

publication criteria?

Reviewer #1: Yes

3. Has the statistical analysis been performed appropriately and rigorously?

Reviewer #1: Yes

4. Have the authors made all data underlying the findings in their manuscript fully available (please refer to the Data Availability Statement at the start of the manuscript PDF file)?

Reviewer #1: No

5. Is the manuscript presented in an intelligible fashion and written in standard English?

Reviewer #1: Yes

Reviewer #1: (No Response)

**Do you want your identity to be public for this peer review?** For information about this choice, including consent withdrawal, please see our Privacy Policy

Reviewer #1: No
